# Thermochromic Smart Windows Assisted by Photothermal Nanomaterials

**DOI:** 10.3390/nano12213865

**Published:** 2022-11-02

**Authors:** Yong Zhao, Haining Ji, Mingying Lu, Jundong Tao, Yangyong Ou, Yi Wang, Yongxing Chen, Yan Huang, Junlong Wang, Yuliang Mao

**Affiliations:** School of Physics and Optoelectronics, Xiangtan University, Xiangtan 411105, China

**Keywords:** thermochromic, smart window, photothermal materials, phase change

## Abstract

Thermochromic smart windows are optical devices that can regulate their optical properties actively in response to external temperature changes. Due to their simple structures and as they do not require other additional energy supply devices, they have great potential in building energy-saving. However, conventional thermochromic smart windows generally have problems with high response temperatures and low response rates. Owing to their great effect in photothermal conversion, photothermal materials are often used in smart windows to assist phase transition so that they can quickly achieve the dual regulation of light and heat at room temperature. Based on this, research progress on the phase transition of photothermal material-assisted thermochromic smart windows is summarized. In this paper, the phase transition mechanisms of several thermochromic materials (VO_2_, liquid crystals, and hydrogels) commonly used in the field of smart windows are introduced. Additionally, the applications of carbon-based nanomaterials, noble metal nanoparticles, and semiconductor (metal oxygen/sulfide) nanomaterials in thermochromic smart windows are summarized. The current challenges and solutions are further indicated and future research directions are also proposed.

## 1. Introduction

According to the report published by the China Association of Building Energy Efficiency in 2021, building energy consumption accounted for 46.5% of the country’s total energy in 2018 of China. Among them, 21.7% is used for energy consumption in the building operation stage [1], that is, the energy consumed by terminal equipment, such as heating, refrigeration, ventilation, and lighting in the building. Additionally, the proportion is still increasing. In the environment of a global energy shortage, the increase in building energy consumption has become a weakness that has restricted the development of the Chinese economy. The percentage of energy lost from windows directly or indirectly is 60%. Therefore, scholars have invented a variety of smart windows [2,3,4,5,6,7] to improve the energy-saving effect of building windows.

In recent years, with energy-saving and green development being vigorously promoted, smart windows have attracted much attention due to their outstanding energy-saving effect. The concept of “smart windows” was introduced by Granqvist in the 1980s [8]. They are energy-saving windows that regulate solar radiation by combining a dimming material [9,10,11] with a base material, such as glass. As optical devices, their core is the sensitive material attached to the glass. Under external excitation by light, electromagnetic radiation, and temperature, the sensitive materials will color or fade, thereby changing the windows’ colors and other optical properties so that they can automatically adjust the indoor temperature and light intensity according to the surrounding environment and ultimately achieve the purpose of saving energy consumption.

At present, most typical smart windows require an external power supply or heating device to dynamically tune the optical properties of materials in response to external stimuli. However, these additional devices have significant disadvantages, such as a complex structure, low lifespan, and high energy consumption, which also greatly hinder their commercial utilization. In contrast, solar-driven thermochromic smart windows can adjust the optical properties actively through the strong absorption of solar energy by photothermal materials, achieving a dual response to light and temperature. The features of low cost, simple structure, and low energy consumption are more conducive to large-scale applications in the future.

Smart windows assisted by photothermal materials are extremely important for the development of smart windows. However, systematic summaries of this aspect are relatively rare. Therefore, in this review, several thermochromic smart windows with different substrates are presented, the auxiliary effects of different kinds of photothermal materials on the phase change of smart windows are reported, and, finally, future research trends are proposed. This will provide references and suggestions for overcoming the problems of a high response temperature and low response rate of traditional thermochromic smart windows.

## 2. Thermochromic Materials for Smart Windows

There are many kinds of thermochromic materials that play an important role as substrates for smart windows. However, only those materials with phase change temperatures adjustable to room temperature (about 28 °C) can be used for building energy efficiency [12]. Therefore, the most widely used photothermal materials for smart windows are vanadium dioxide (VO_2_), thermochromic hydrogels, and liquid crystals, as shown in Figure 1.

Vanadium dioxide (VO_2_) is a kind of metal-insulator phase transition (MIT) material, which is extensively utilized in thermochromic smart windows [13,14,15,16]. Its phase transition temperature is close to room temperature at 68 °C [17]. As the temperature rises, VO_2_ will undergo a phase transition from a semiconductor to a metal (Figure 1a), with a sudden change in infrared transmittance while keeping the visible transmittance unchanged [18]. Although VO_2_ films have great practical value, which are used in smart windows [19], photoelectric switches [20,21], and infrared stealth [22,23,24], the high phase transition temperature and poor optical properties are still problems to be solved.

Liquid Crystal (LC) is a special material between traditional liquids and crystalline solids. Its optical, electrical, and mechanical characteristics are anisotropic, and it can also be employed in smart windows [25,26,27,28]. From the molecular point of view, the thermochromism of Liquid Crystals can be achieved by anisotropic molecules responding to temperature and adjusting their orientation (Figure 1b). For example, the molecules inside the polymer network Liquid Crystals (PNLC) are arranged in order at low temperatures, and the refractive index of the liquid crystal and the polymer are the same, showing a highly transparent state. As the temperature rises, the liquid crystals transform into a blurred state due to the disorder of molecular orientation, and the transmittance, including near-infrared light, can change by more than 20%.

Hydrogels are both hydrophilic and hydrophobic and have been commonly employed in the field of smart windows [29,30,31,32]. Thermochromic hydrogels, such as poly(N-isopropylacrylamide) (PNIPAm) and hydroxypropyl methyl cellulose (HPMC), have a lower critical solution temperature (LCST) of around 32 °C to 40 °C. When the temperature is below the LCST, hydrophilic groups in the hydrogel form hydrogen bonds with water molecules, presenting a highly transparent state. Above the LCST, the hydrogen bonds are broken and polymers aggregate, resulting in a significant decrease in transmittance in the entire spectral range [33] (Figure 1c). Thermochromic hydrogels can reversibly change transparency with an increase in temperature, making them ideal materials for smart windows.

## 3. Common Photothermal Conversion Materials

Photothermal and thermochromic materials are often combined for sunlight-driven thermochromic smart windows. The strong light-absorption characteristics or localized surface plasmon resonance effects [34,35] of photothermal materials can convert solar energy into heat energy, increase the surrounding temperature, and achieve the purpose of assisting the phase transition.

### 3.1. Carbon-Based Nanomaterials

Carbon-based nanomaterials are a new type of photothermal material developed in recent years, such as carbon dots [36,37], graphene, and graphene oxide [38,39,40]. They have strong absorption in visible and near-infrared light and can transform solar energy into heat energy rapidly owing to their unique structures. Therefore, they show a wide range of applications in smart windows [41,42,43,44,45,46], photocatalysis [47], and tumor-targeted therapy [48].

Graphene oxide (GO) is a two-dimensional nanomaterial with a single atomic thickness obtained by the oxidation and dispersion of natural graphite. Under sunlight, most of the energy photons in visible light can be absorbed by electrons, so they are excited. When the excited-state electrons fall back to the ground state, they release heat and raise the local temperature, producing a significant photothermal effect.

The excellent photothermal performance of GO was verified by Kim [49] through the control experiment. In the experiment, the aqueous graphene oxide (GO) could convert solar energy into heat energy under sunlight, raising the solution temperature by 5 °C in 8 min, while the temperature of pure water had little change (Figure 2a).

Based on this, GO/PNIPAm hydrogels were successfully prepared [41]. The GO dispersed in the PNIPAm hydrogels allowed the originally transparent films to reach the LCST at room temperature (30 °C) and complete the phase change in 2 min (Figure 2b). It is worth noting that similar research was conducted by Zhu [42]. The results showed that the heating effect and rate were positively correlated with the GO concentration, but had no significant effect on the conversion temperature of the composite hydrogel.

In order to obtain a wider phase transition temperature range, Lee [43] et al. prepared multicomponent copolymers with a composition gradient. Then, NIPAm-co-NDEAm-co-VP(PNDV) hydrogel and graphene oxide (GO) composites were sealed in double-layer glass to assemble a new type of embedded glass. Compared with ordinary hydrogel glass, this novel glass achieved step-by-step solar control of the smart window in response to sunlight intensity (Figure 2c), providing users with more choices.

Although hydrogels have been widely studied in the field of smart windows, the poor mechanical properties of traditional hydrogels greatly limit their applications. Near-infrared light (NIR)-responsive (L-PNIPAm/GO) hydrogels were prepared [44] by the in situ polymerization of N-isopropylacrylamide (NIPAm), in which laponite (L) was the crosslinking agent and graphene oxide (GO) was the photothermal conversion agent. The addition of GO not only improved the hydrogels’ photothermal sensitivity, but also raised the system’s cross-linking point, resulting in a more flawless structure. Through the characterization of multiple samples, it was found that the prepared L-PNIPAm/GO-12.0-0.21 hydrogels had high strength and good toughness when the contents of laponite and GO were 12.0% and 0.21% of the mass of NIPAm. Under NIR irradiation, it could be heated from 20.3 °C to 48.5 °C within 300 s, with good temperature sensitivity and photothermal effect. Thus, it has great potential in the fields of smart windows and light-controlled switches.

Graphene oxide can also be an effective auxiliary in the phase transition of VO_2_. A series of tungsten-doped vanadium dioxide/graphene composites was prepared by a hydrothermal method using graphene oxide powders (GO), vanadium pentoxide, oxalic acid, and ammonium tungstate as raw materials [45]. Due to the doping of tungsten and the addition of GO, VO_2_ is uniformly loaded on the surface of graphene as spherical particles, the particle size is significantly reduced, and the agglomeration is improved. When the atomic percentage of tungsten was 2.5% and the content of GO was 4%, the phase transition temperature of VO_2_ was reduced from 66.0 °C to 32.2 °C. In addition, the thermal conductivity reached the maximum value of 16.341 W/(m·K), which effectively improved the heat dissipation effect of the coating. Such an improvement in properties allows it to better meet the required functionality of an intelligent thermal insulation coating.

There are many varieties of carbon-based nanomaterials, and different structures afford them different characteristics. Among them, carbon dots (CDS) are identical to GO in many ways, including the chemical structure and physical qualities. As a new zero-dimensional carbon nanomaterial with a size of less than 10 nm, it has good photothermal properties. A light-responsive material was reported by Shen [46], which was obtained by incorporating carbon dots (CDs) into hydroxypropylmethyl cellulose (HPMC). Through a series of studies, it was found that the ultrasmall size of CDs guaranteed the considerable transparency of CDs/HPMC and enhanced light absorption. Moreover, chloride-modified CDs (Cl-CDs) can greatly shorten the response time and lower the LCST, which also provides a new idea to improve the material’s function by regulating the surface characteristics of CDs.

### 3.2. Noble Metal Nanoparticles

Noble metal nanoparticles are the most researched near-infrared photothermal materials, such as Au [50,51], Ag [52], Pd [53,54], etc. Their photothermal effect is mainly derived from the strong localized surface plasmon resonance effect of nanoparticles, which is a unique phenomenon occurring in metal structures. When the frequency of the incident light matches the eigenfrequency of the free electrons in the metal, the electrons will be collectively excited and resonated. Vibrating electrons will convert kinetic energy into heat energy due to the damping effect, thus increasing the local temperature [55].

The most common noble metal nanomaterials exhibiting the plasmon resonance effect are gold nanocrystals (AuNCs). Sunlight-responsive thin HPMC/AuNCs films, combining the thermochromic properties of HPMC and the photothermal effect of AuNCs, have been prepared successfully via a simple and low-cost method [56]. Under laser irradiation, the temperature of HPMC films containing AuNRs was observed to increase rapidly and reach a steady state within 80 s. Their steady-state temperature was close to 69 °C, however, the temperature of the film without AuNRs only increased by 1 °C under the same conditions (Figure 3a). Additionally, the switching temperature and colors of the film could be flexibly controlled by adjusting the concentration of AuNCs and the type of thermochromic material (Figure 3b).

Compared with gold nanospheres (AuNPs), aspheric AuNRs are anisotropic and exhibit different LSPR properties. Based on this, novel, random-filled composite nanohydrogels were obtained by an in situ reduction method. The nanohydrogels were mainly composed of AuNRs and PNIPAm [59]. It was found that there were two completely separated plasmon resonance absorption peaks on the absorption spectra of AuNRs. With an increase in temperature, the transverse LSPR peaks altered slightly, while the longitudinal LSPR peak had a strong red shift. Additionally, the amplitude of the resonance enhancement increased with an increase in the length-to-diameter ratio of AuNRs.

Furthermore, noble metal nanoparticles can be improved to exert their photothermal effects more efficiently by building particular structures or coatings. Wei et al. developed a new strategy for thermal-responsive PNIPAm–acrylic/Ag NR hybrid hydrogels [57]. The hydrogels had full-wavelength thermal management functions, which can be used for smart windows. Initially, AgNRs are coated with polymers and point in different directions. When the temperature rises above the LCST, the hydrogel structure shrinks and deforms, causing the AgNRs to stand up like flowers (Figure 3d,e), and their IR emissivity slightly increases from 0.947 to 0.958. In addition to this, the hydrogels can be fully transparent at room temperature, with a high solar modulation capability (Δ*T_sol_* = 59.24%) and ultrahigh luminous transmission (*T_lum_* = 61.36%) after phase transition. Therefore, they can be used for solar radiation modulation between the visible, near-infrared (NIR), and mid-infrared (MIR) regions.

The combination of noble metal nanoparticles and VO_2_ also has significant application potential in the field of smart windows. Periodic arrays of Ag nanodiscs on VO_2_ films were fabricated by Shu [58], and their LSPR effect brought afforded absorption to the films in the visible regions. Moreover, by tuning the diameter of the Ag nanodiscs and spatial periodicity of the arrays, various colors could be actively adjusted in the whole visible spectrum (Figure 3c), greatly increasing the color diversity of the VO_2_ films.

Moreover, a vanadium dioxide (VO_2_) and gold nanotriangle composite sandwich structure (VO_2_/Au/VO_2_) was prepared via nanosphere lithography [60]. The gold nanotriangles in the films exhibited periodic arrangement, and their sharp edges and tips promoted the occurrence of LSPR. This sandwich architecture may provide enlightenment for the design of temperature-sensitive smart windows.

With the in-depth study of nanoscience, it has been found that composites prepared by mixing two or more noble metal nanoparticles often combine their respective advantages and have superior physicochemical properties than single nanoparticles. Au@AgNR@PNIPAM microgels, with Au@AgNR as the core and cross-linked PNIPAM as the shell, were prepared via seeded precipitation polymerization [61]. The combination of the two metals not only improves the antioxidant capacity of AgNRs, but also overcomes the problem of the weak enhancement of the electric field on the surface of AuNRs.

Although noble metal materials, such as Au, Ag, and Pd, have an excellent theoretical basis in the fields of catalysis [62,63], photothermal devices [64], biomedicine [65], etc., their scarce reserves and high prices seriously limit their large-scale promotion. This is also the greatest problem in the practical application of noble metal nanomaterials.

### 3.3. Semiconductor Nanomaterials

Due to the wide energy gap, conventional semiconductor materials need to absorb UV light with higher energy to excite the electrons and release heat in the process of falling back to the ground state. However, with the study of more kinds of semiconductor materials, it has been found that localized surface plasmon resonance (LSPR) exists not only in noble metals, but also in semiconductor materials with appreciable free carrier density, such as tin-doped indium oxide (ITO) [66,67], copper sulfide (Cu_x_S) [68,69], titanium nitride (TiN) [70,71], etc. Compared with conventional noble metals, semiconductors can exhibit LSPR in both the ultraviolet-visible (UV-vis) and near–mid-infrared (IR) spectral regions, which significantly extends the light absorption range.

#### 3.3.1. Metal Oxide

Visible light transmittance (*T_lum_*) is an important indicator for smart windows. Therefore, when selecting photothermal materials, their impact on transparency should also be given priority. Antimony-doped tin dioxide (ATO) is a transparent N-type semiconductor with special photoelectric effects and excellent light absorption properties. Its UV absorption is caused by the material’s inherent wide bandgap, and the NIR absorption is attributed to the localized surface plasmon resonance (LSPR) induced by N-type doping [72].

A novel, fully autonomous light-driven thermochromic material was proposed, consisting of hybridizing poly-N-isopropylacrylamide (PNIPAM) hydrogel and antimony–tin oxide (ATO) [72]. Even when the outdoor temperature is far lower than the transition temperature, PNIPAM/ATO (PATO) composite hydrogels can also use ATO as a nanoheater to induce the optical switching of the hydrogel under sunlight. Furthermore, by increasing the Sb dopant content, the free electron concentration can be increased and the NIR absorption efficiency can be enhanced, which means that the ATO can convert more solar energy into heat. It is worth noting that the 10 atom % Sb-doped PATO not only showed the best response speed and solar modulation ability, but also achieved a subtle balance between thermal comfort and visuality. On this basis, Xu [73] prepared supramolecular nanocomposite hydrogel films by integrating ethylene glycol-modified pillar arene (EGP5) and antimony–tin oxide (ATO) nanoparticles (Figure 4a). The obtained films displayed good initial luminous transmission (77.2%) and excellent solar modulation ability (56.1%) under 100 mW/cm^2^ xenon lamp irradiation owing to the thermal responsiveness of EGP5 and plasma heating induced by NIR absorption of ATO (Figure 4b). At the same time, the host–guest interaction between EGP5 and pyridinium units efficiently avoids the collapse of and damage to the hydrogel structure, resulting in a film that possessed high repeatability and durability.

ATO can also be used in other types of smart windows. Novel smart window films were successfully prepared by doping ATO nanoparticles into polymer-stabilized Liquid Crystal (PSLC) [75]. These films had a wider waveband modulation function, covering the visible and infrared regions (380–5500 nm), and the transmittance could be altered in the visible region from a highly transparent (78.5%) state to a strong light-scattering (10%) state. In addition, owing to the LSPR of ATO nanoparticles, up to 80.7% of the IR light could be effectively shielded.

Tin-doped indium oxide (ITO) is a promising NIR shielding material similar to ATO. A flexible, multi-responsive material was invented by creating a compatible interface between ITO nanocrystals and polar syrup. Polar syrup is a Liquid Crystal material that initially combines the good mechanical strength of PDLC with the high transparency of PSLC [76]. The Liquid Crystal molecules in it can change from homeotropically oriented to a focal–conic texture in response to the electric field and temperature, achieving a rapid transition from the transparent state (78%) to the opaque state (1.5%).

Cesium tungsten bronze (Cs_x_WO_3_) is an excellent photothermal material with high absorption in both the UV and NIR regions. A spectrally selective smart window using Cs_x_WO_3_ as the photothermal component and PAM hydrogel matrix embedded with thermally responsive PNIPAM microgels as the photocontrol switch was designed [74] (Figure 4c,d). Under solar irradiation, the steady-state surface temperature of the Cs_x_WO_3_/PAM-PNIPAM window can remain at 47 °C. It can shield about 96.2% of the near-infrared radiation while ensuring good visible light transmission, so that the room temperature is maintained at about 25°C.

Moreover, a novel roll-to-roll fabrication of multi-responsive flexible Liquid Crystal material was presented [77]. It can not only control the transmission of visible light by the stimulus of voltage, heat, or NIR light, but also shield nearly 95% of the NIR radiation in the 800–2500 nm band. Excellent properties of the films were achieved by forming a compatible poly(vinylpyrrolidone) (PVP) tuning layer between Cs_x_WO_3_ nanorods (NRs) and polymeric syrup containing liquid crystals. It also offers a wide working temperature range, improved flexibility, long-term stability, high mechanical strength, and large-area processability. These studies on the combination of Cs_x_WO_3_ and smart windows point toward a new route for the development of multifunctional and energy-efficient smart windows.

#### 3.3.2. Metal Sulfide

Metal sulfides, such as PbS [78], MoS_2_ [79], and Cu_7_S_4_ [80,81], have the advantages of low cost, good photothermal stability, and shape-controlled particle sizes. These features are beneficial to reduce the cost and the difficulty of the large-scale preparation of smart window films.

PbS is a narrow band gap semiconductor with a bandwidth of 0.4 eV. Its absorption band covers the main area of solar energy, making it an excellent light-absorbing coating. The PVA-PbS-VO_2_ (M) composite film was fabricated by Qi [82] using the casting film method. It can effectively absorb sunlight from 0.3 to 2.5 μm through the interaction of solar photons with PbS nanostructured phonons and convert it into heat energy so that VO_2_ undergoes a phase transition at room temperature.

Molybdenum disulfide (MoS_2_) is a semiconductor material with a two-dimensional layered structure similar to graphene, which has strong absorption of near-infrared light. The MoS_2_/PNIPAM composite hydrogels were prepared by incorporating chemically exfoliated MoS_2_ nanosheets into poly(N-isopropylacrylamide) (PNIPAM) hydrogels [83]. It not only inherits the good thermal-responsive properties from the PNIPAM hydrogel, but also significantly shortens the response time. However, due to the black color of MoS_2_, the coordination of the MoS_2_ content and the transparency of smart windows will be the focus of future research.

Cu_7_S_4_ nanocrystals are a p-type semiconductor nanomaterial with an indirect bandgap. Because of their localized surface plasmon resonance (LSPR) properties in the NIR region, they have broad application prospects in solar cells, environmental pollution treatment, and photothermal treatment. Cu_7_S_4_/poly (N-isopropylacrylamide) (PNIPAM) composite films with fast optical/thermal response were prepared by Zhu [84]. When the temperature exceeds 32 °C, the smart window can quickly change from transparent to opaque within 3 min based on the wide absorption of sunlight and the high photothermal conversion efficiency of Cu_7_S_4_ nanoparticles. As long as the sunlight is strong enough, this smart window can even raise the indoor temperature by absorbing and converting sunlight in cold weather.

Through research on the shape-controlled synthesis of Cu_7_S_4_, it was found that there was an obvious size dependence on its localized surface plasmon resonance [85]. That is, by increasing the crystal size, the maximum resonance absorption peak shifted to red and the absorption intensity also increased. This research is beneficial to further improve the light absorption properties of Cu_7_S_4_ nanomaterials.

### 3.4. Others

Transition metal nitrides, such as titanium nitride, are versatile metal–ceramic materials with high-temperature durability, chemical stability, corrosion resistance, electrical conductivity, and also localized surface plasmon resonance properties [86,87,88]. It can replace noble metal plasmas to generate LSPR and promote material phase changes under light conditions, resulting in lower-cost thermochromic materials that are available at room temperature.

Vanadium dioxide/titanium nitride (VO_2_/TiN) smart coatings were produced by Hao [89] via coating TiN nanoparticles with pure monoclinic-phase (M-phase) VO_2_ films. These coatings can control the infrared (IR) radiation dynamically, depending on the ambient temperature and light intensity. The photothermal conversion effect of TiN nanomaterials will induce VO_2_ to change from a monoclinic phase to a tetragonal phase under strong light or higher temperatures (28 °C), which can effectively block IR radiation by nearly 70%; conversely, these coatings show good IR transparency at 20 °C (Figure 5a). In addition, the visible light transmittance of these coatings at 2000 nm is 51%, and the infrared conversion efficiency is 48%. These unique features result in VO_2_-TiN having great potential in the field of energy-saving windows.

Not only can inorganic materials be used as photothermal materials, but also organic polymers, such as polydopamine nanoparticles (PDAPs) [90,91]. Compared with conventional materials, PDAPs have excellent near-infrared responsivity and high thermal conductivity, as well as good biocompatibility and oxidation resistance. They also have the characteristics of easy preparation, low cost, and high yields.

Based on this, a new, fully autonomous thermochromic smart window containing PDAPs was constructed via a one-step in situ polymerization method (Figure 5b) [92]. It was shown that the addition of PDAPs not only ameliorated the microstructure and mechanical properties of the hydrogel, but also improved its photothermal conversion ability. It could also trigger the smart window to quickly adjust the indoor light and temperature, even if the ambient temperature was below the LCST of the hydrogel. This bold attempt provides a new idea for the research into multifunctional thermochromic smart windows.

## 4. Conclusions and Perspectives

In recent years, great progress has been made in the research into the phase transition of photothermal material-assisted thermochromic smart windows. In this review, an overview of several thermochromic materials from VO_2_, Liquid Crystal, and hydrogel is provided. The photothermal conversion efficiency of carbon-based nanomaterials, noble metal nanoparticles, and semiconductor (metal-oxygen/sulfide) nanomaterials (Table 1), as well as their applications in thermochromic smart windows (Table 2), were summarized. The current challenges and solutions are further pointed out and future research directions are also proposed.

The use of thermochromic smart windows has been discussed for many decades. Typically, the high response temperature, poor solar modulation, and slow optical switching are considered as the main drawbacks of traditional thermochromic smart windows. Therefore, researchers are trying to introduce photothermal materials into smart windows and use the photothermal conversion effect to assist the phase transition of smart windows. This can not only achieve the dual regulation of light and heat at room temperature, but also greatly shorten the response time and drive the development of smart windows toward low energy consumption and high universality. The outlooks and prospects for future research on the phase transition of photothermal material-assisted thermochromic smart windows are suggested as follows:(1)Improvement of morphology (size and shape) of photothermal materials.

Owing to the size and shape dependence of photothermal nanomaterials, the optical absorption characteristics can be effectively adjusted by changing the particle size or adopting different structures, such as cages, core–shells, and so on. For example, in Table 1, the photothermal efficiency of a Cu_7_S_4_ Nano Superlattice is significantly higher than that of Cu_7_S_4_ NPs [99]. In addition, photothermal nanoparticles of different sizes can also be mixed together so that their extinction spectrum can be better matched with the solar radiation spectrum, and as much solar energy can be harvested as possible.

(2)Composites of different kinds of photothermal materials.

Based on the absorption spectrum characteristics of various photothermal materials, composite photothermal materials could be incorporated into the smart window system. Au@Cu_7_S_4_ yolk–shell NPs [100] not only retain the respective absorption peaks of Au and Cu_7_S_4_ in UV-Vis-NIR spectra, but also show significantly enhanced absorption in the near-infrared region. Additionally, the photothermal conversion efficiency is also higher, which shows that the total is greater than the sum of parts.

(3)Construction of bionic anti-reflection structure.

In addition to improving the conversion efficiency of photothermal materials, the solar absorption efficiency is also an important parameter. In the field of solar cells, bionic anti-reflection periodic structures, such as butterfly wings and moth eyes, are often used on the surface of solar panels [103]. These rough nanostructures can effectively eliminate reflection and help them absorb more sunlight than smooth surfaces. Similarly, this structure is also applicable to smart windows, which will significantly shorten the response time.

## Figures and Tables

**Figure 1 nanomaterials-12-03865-f001:**
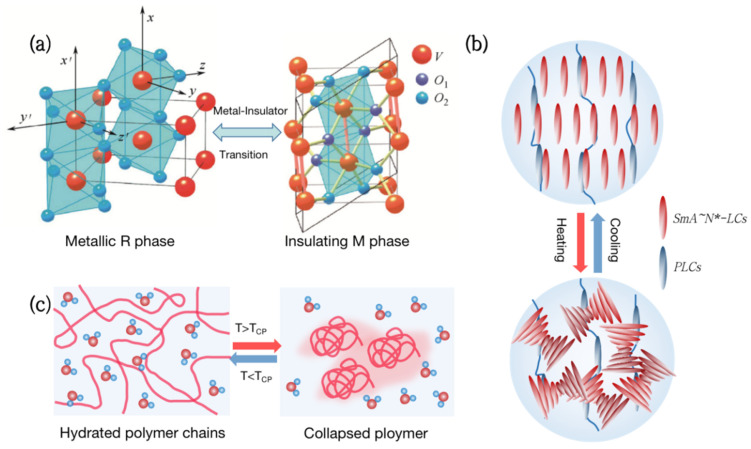
Types of thermochromic smart windows. (**a**) Metal-insulator phase transition (MIT) of vanadium dioxide (VO_2_). (**b**) Liquid crystal changes transmittance by adjusting molecular orientation in response to temperature. (**c**) Thermochromic hydrogels change transparency reversibly with temperature.

**Figure 2 nanomaterials-12-03865-f002:**
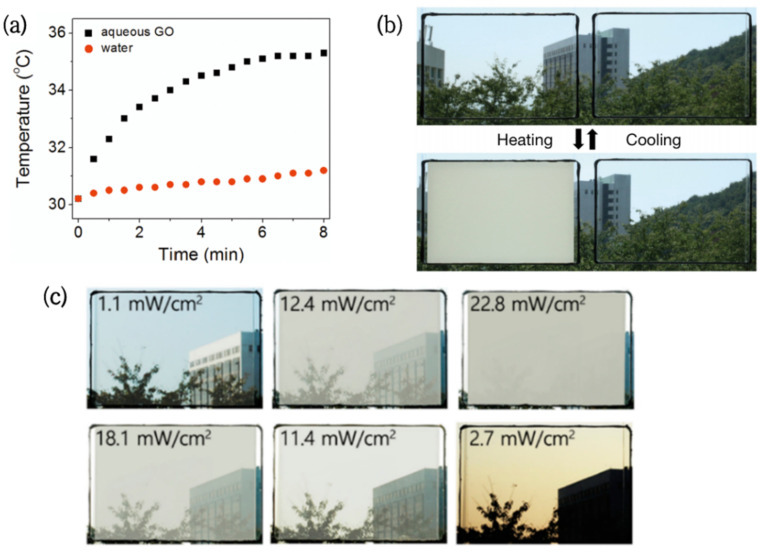
(**a**) Temperature change of pure water and 2.5 mg/mL GO aqueous solution under visible light irradiation [49]. (**b**) Photo of the transparency transition of PNIPAMm/GO hydrogel window (left) and PNIPAm hydrogel window (right) under sunlight [41]. (**c**) Photo of the PNDV/GO hydrogel window transmittance gradient change with light intensity [43].

**Figure 3 nanomaterials-12-03865-f003:**
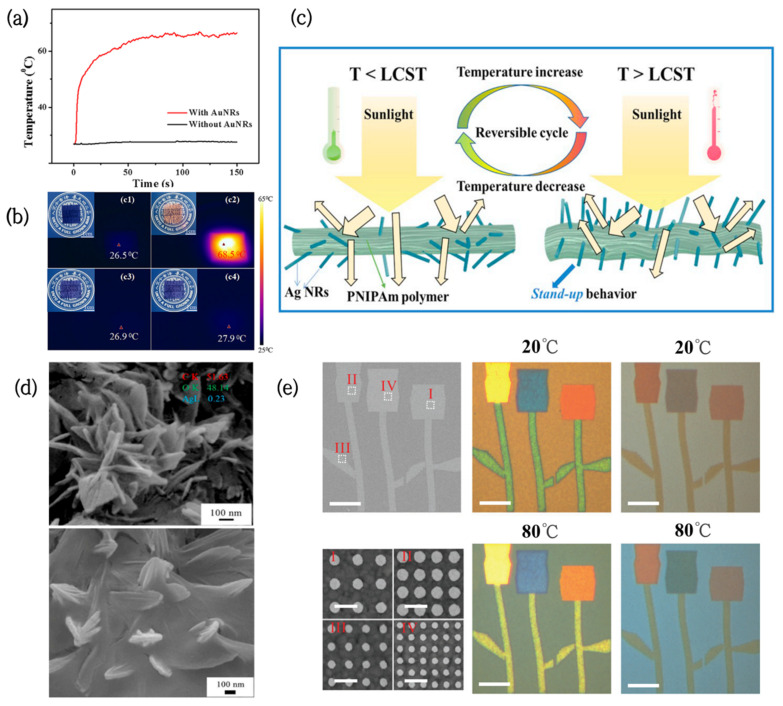
(**a**) Temperature rise curves of PVA/dye films with and without AuNRs [56]. (**b**) Digital photographs and infrared photographs of PVA/dye/AuNRs films and PVA/dye films before and after laser irradiation [56]. (**c**) Schematic representation of the structural shrinkage and deformation of PNIPAm acrylic/AgNRs when the temperature is higher than LCST [57]. (**d**) SEM images of PNIPAm-acrylic acid/AgNRs hydrogels at different temperatures [57]. (**e**) Color patterns constructed from arrays of Ag nanodiscs on silica films at different temperatures [58].

**Figure 4 nanomaterials-12-03865-f004:**
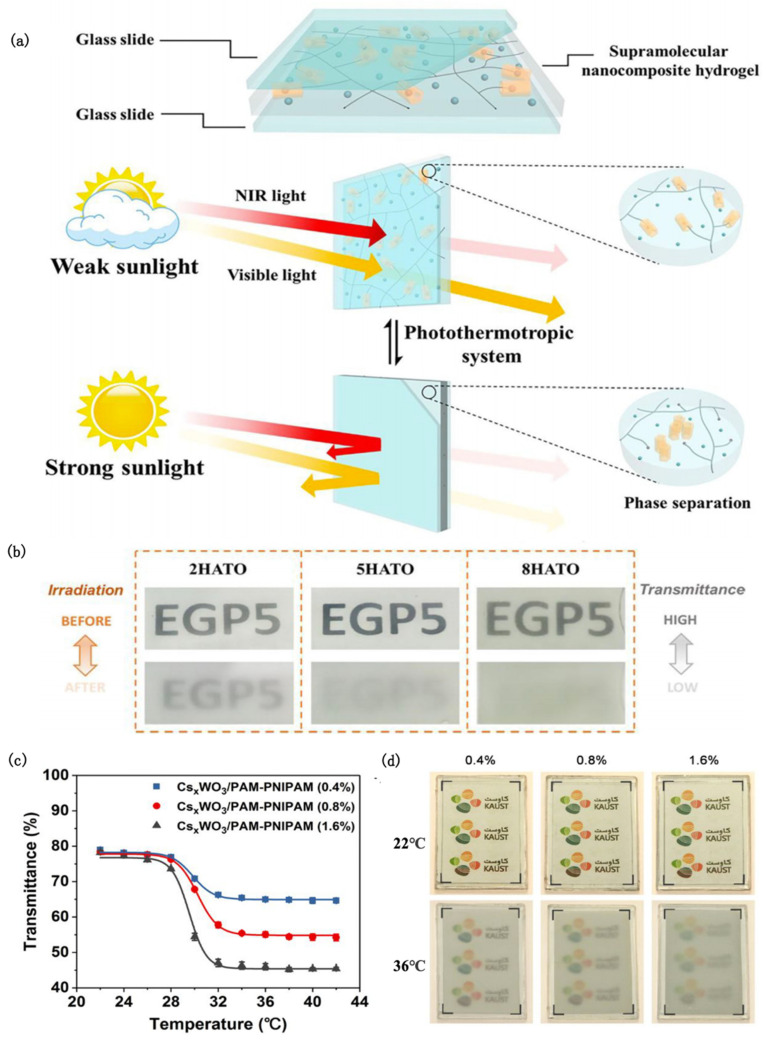
(**a**) Schematic diagram of the chemical structure and working principle of the photochromic supramolecular hydrogel smart window [73]. (**b**) Photographs of ATO composite hydrogel films with EGP5 contents of 2, 5, and 8 mol%, respectively, before and after irradiation at 100 mW/cm^2^ for 10 min [73]. (**c**) Transmittance variation of Cs_x_WO_3_/PAM-PNIPAM film with temperature at 550 nm wavelength [74]. (**d**) Transparency changes of Cs_x_WO_3_/PAM-PNIPAM films with different PNIPAM concentrations at 22 °C (above) and 36 °C (below) [74].

**Figure 5 nanomaterials-12-03865-f005:**
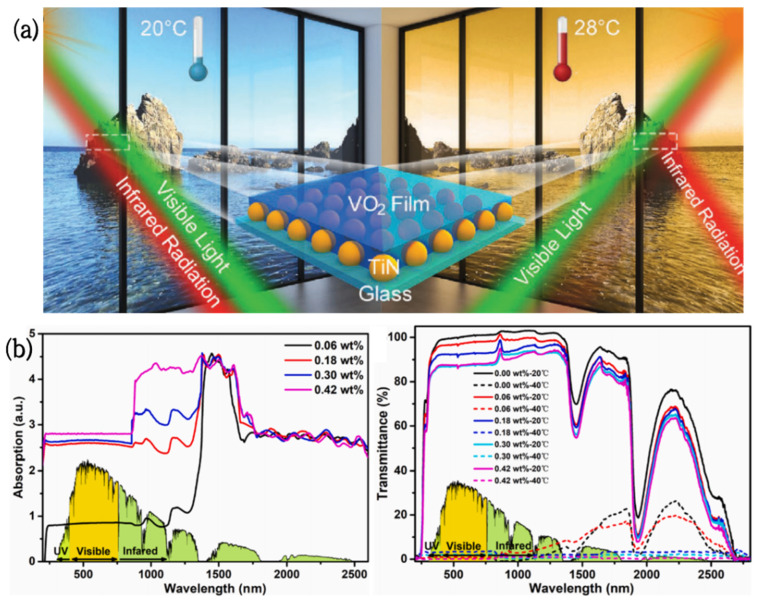
(**a**) Schematic diagram of a VO_2_/TiN smart window [86]. (**b**) UV-Vis-NIR absorption and transmission spectra of PNIPAm hydrogels containing different levels of PDAPs at 20 °C and 40 °C [87].

**Table 1 nanomaterials-12-03865-t001:** Conversion efficiency of photothermal materials.

Type	Power of Excitation Light Source	Efficiency	Reference
GO NSs	Simulated sunlight (0.1 W/cm^2^)	63.2%	[93]
CDs	AM1.5 Standard sunlight (0.1 W/cm^2^)	23.46%	[46]
Au NRs	808 nm laser (1.27 W/cm^2^)	39.2%	[94]
Au STs	808 nm laser (2 W/cm^2^)	42%	[95]
ATO Hollow ball	1064 nm laser (-)	30.69%	[96]
Cs_0.33_WO_3_	980 nm laser (1.66 W/cm^2^)	23.1%	[97]
Cu_9_S_5_ NCs	980 nm laser (0.51 W/cm^2^)	25.7%	[98]
Cu_7_S_4_ NPs	808 nm NIR laser (1 W/cm^2^)	56.32%	[99]
Cu_7_S_4_ Nano Superlattice	808 nm NIR laser (1 W/cm^2^)	65.7%	[99]
Au@Cu_7_S_4_ Yolk–shell NPs	980 nm laser (0.51 W/cm^2^)	63%	[100]
TiN	808 nm NIR laser (1.5 W/cm^2^)	38.6%	[101]
PDAPs	808 nm NIR laser (2 W/cm^2^)	40%	[102]

**Table 2 nanomaterials-12-03865-t002:** Performance comparison of thermochromic smart window.

Thermochromic Material	Photothermal Material	*T_lum,l_*	*T_lum,h_*	Δ_Tsol_	*T_C_*	Reference
HPMC	-	98.94%	89.32%	6.52%	40°C	[46]
PNIPAM	-	73.2%	48.58%	26.88%	31.4°C	[58]
VO_2_	-	47%	52%	ΔT_NIR_ = 51%	68°C	[86]
PNIPAM	GO	97.8%	3.0%	-	32.5°C	[41]
HPMC	CDs	82.11%	3.50%	65.55%	45°C	[46]
HPMC	Au NCs	85.4%	3.5%	62.6%	40°C	[56]
VO_2_	Au	74.1%	66.6%	13.2%	50°C	[60]
PNIPAM	Ag NRs	61.36%	3.43%	59.24%	32.9°C	[58]
PNIPAM (10%Sb)	ATO NPs	83.0%	55.3%	35.7%	33.6°C	[72]
PNIPAM	ATO NPs	77.2%	18.6%	56.1%	32.1°C	[73]
PSLC	ATO NPs	78.5%	10%	-	-	[74]
PDLC-PSLC	ITO NCs	78%	1.5%	-	28.3°C	[75]
PAM-PNIPAM	Cs_x_WO_3_	78.2%	45.3%	-	30°C	[76]
PT-LCs	Cs_x_WO_3_ NRs	67%	1.5%	-	30.5°C	[77]
PNIPAM	Cu_7_S_4_ NPs	79%	41.1%	31.2%	32°C	[84]
VO_2_	TiN NPs	49%	51%	ΔT_NIR_ = 48%	28°C	[85]
PNIPAM	PDAPs	92.58%	3.41%	88.9%	33°C	[92]

## Data Availability

Not applicable.

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
