# Peer review of "Thermochromic Smart Windows Assisted by Photothermal Nanomaterials"

_nanomaterials, 2022, doi:10.3390/nano12213865_

Round 1

Reviewer 1 Report

The paper reviews thermochromic window using photothermal nanomaterials. However, this paper has many points to be improved. Please address the following comments.

1.         The introduction is too short. Readers cannot understand why review is necessary for this field. Please explain more about the value, strategy, contribution of this review.

2.         The number of the references are also too small. There are many papers related to this field. At least 100 references are required.

3.         Please improve the structure of the review paper. For example, the section 2 is entitled as “Types of thermochromic smart windows”. However, there are explanation of some materials. After that, “3 Common photothermal conversion materials” is placed.

4.         In the conclusion section, the authors give suggestion of (1) ~ (3). However, these are typical opinions. Please discuss deeper related to the mentioned references.

Author Response

Dear Editor and Referees:

Thanks for your letter and for the Referees’ comments concerning our manuscript entitled “Thermochromic Smart Windows Assisted by Photothermal Nanomaterials” (Manuscript ID: nanomaterials-1989252). We deeply appreciate the time and effort you’ve spent on reviewing our manuscript. These comments are really thoughtful and very helpful for revising and improving our paper, as well as the important guiding significance to our research. During the last days, we have referred to literatures, checked spelling and studied your comments carefully, and we have completed the modifications according to the referees’ comments.

The Referees’ comments and the responses to the comments are as follows:

Reviewer: 1#

The paper reviews thermochromic window using photothermal nanomaterials. However, this paper has many points to be improved. Please address the following comments.

Response:

We thank the Referee for the positive comments on our work. According to the Referee’s suggestion, we have supplemented the introduction, increased the number of references, and modified the conclusions. The modified part is highlighted.

Comment 1:

The introduction is too short. Readers cannot understand why review is necessary for this field. Please explain more about the value, strategy, contribution of this review.

Response:

Thanks for Referee’s kind suggestion. It is really true as the Referee suggested that the value, strategy, contribution of this review should be added. According to the Referee’s suggestion we have added this part in the revised manuscript. The modified is:

In recent years, with energy-saving and green development being vigorously promoted in recent years, smart windows have attracted much attention due to their outstanding energy-saving effect. The concept of "smart windows" was introduced by Granqvist in the 1980s[8]. It is an energy-saving window that regulates solar radiation by combining a dimming material[9-11] with a base material such as the glass. As an optical device, its core is the sensitive material attached to the glass. Under external excitation of light, electromagnetic radiation and temperature, the sensitive materials will color or fade thereby changing its own color and other optical properties, so that they can automatically adjust indoor temperature and light intensity according to the surrounding environment, and ultimately achieve the purpose of saving energy consumption.

At present, most typical smart windows require the external power supply or heating device to dynamically tune the optical properties of materials in response to external stimuli. However, these additional devices have significant disadvantages such as complex structure, low lifespan, and high energy consumption, which also greatly hinder their commercial utilization. In contrast, solar-driven thermochromic smart windows can adjust the optical properties actively through the strong absorption of solar energy by photothermal materials, achieving a dual response to light and temperature. The features of low cost, simple structure and low energy consumption are more conducive to the large-scale application in the future.

The smart window assisted by photothermal materials is extremely important for the development of smart windows. However, the systematical summary on this aspect is relatively rare. Therefore, in this review, several thermochromic smart windows with different substrates are presented, the auxiliary effect of different kinds of photothermal materials on the phase change of smart windows are reported, and finally the future research trends are prospected. This will provide references and suggestions for overcoming the problems of high response temperature and low response rate of traditional thermochromic smart windows.

Comment 2:

The number of the references are also too small. There are many papers related to this field. At least 100 references are required.

Response:

Thanks for Referee’s kind suggestion. According to the Referee’s suggestion, we have cited some literature published in recent years at appropriate locations, and the total number of references is 103.

(6) Luo, L.; Liang, Y.; Feng, Y. Recent Progress on Preparation Strategies of Liquid Crystal Smart Windows. Crystals 2022, 12, 1426.

(7) Zhou, Y.; Dong, X.; Mi, Y.; Fan, F.; Xu, Q.; Zhao, H.; Wang, S.; Long, Y. Hydrogel smart windows. Journal of Materials Chemistry A 2020, 8, 10007-10025.

(8) Hamberg, I.; Granqvist, C. Evaporated Sn-doped In2O3 films: Basic optical properties and applications to energy‐efficient windows. Journal of Applied Physics 1986, 60, 123-160.

(9) Jing, T. Selective Reflection of Cholesteric Liquid Crystals. Science Insights 2022, 40, 515-517.

(10) Ming, Y.; Sun, Y.; Liu, X. Optical evaluation of a Smart Transparent Insulation Material for window application. Energy Conversion and Management: X 2022, 100315.

(11) Huang, Z.; Chen, S.; Lv, C.; Huang, Y.; Lai, J. Infrared characteristics of VO2 thin films for smart window and laser protection applications. Applied Physics Letters 2012, 101, 191905.

(13) Cui, Y.; Ke, Y.; Liu, C. Thermochromic VO2 for energy-efficient smart windows. Joule 2018, 2, 1707-1746.

(14) Barimah, E.; Boontan, A.; Steenson, D. Infrared optical properties modulation of VO2 thin film fabricated by ultrafast pulsed laser deposition for thermochromic smart window applications. Scientific Reports 2022, 12, 1-10.

(15) Ren, H.; Hassna, O.; Li, J.; Arigong, B. A patterned phase-changing vanadium dioxide film stacking with VO2 nanoparticle matrix for high performance energy-efficient smart window applications. Applied Physics Letters 2021, 118, 051901.

(16) Riapanitra, A.; Asakura, Y.; Yin, S. One-step hydrothermal synthesis and thermochromic properties of chlorine-doped VO2(M) for smart window application. Functional Materials Letters 2020, 13, 1951008.

(25) Meng, W.; Gao, Y.; Hu, X.; Tan, L.; Li, L.; Zhou, G.; Jiang, L. Photothermal Dual Passively Driven Liquid Crystal Smart Window. ACS Applied Materials & Interfaces 2022, 14, 28301-28309.

(26) Meng, C.; Chen, E.; Wang, L.; Tang, S.; Tseng, M.; Guo, J.; Kwok, H. Color-switchable liquid crystal smart window with multi-layered light guiding structures. Optics Express 2019, 27, 13098-13107.

(27) Kragt, A.; Loonen, R.; Broer, D.; Debije, M.; Schenning, A. 'Smart' light-reflective windows based on temperature responsive twisted nematic liquid crystal polymers. Journal of Polymer Science 2021, 59, 1278-1284.

(28) Oh, S.; Nam, S.; Kim, S.; Yoon, T.; Kim, W. Self-regulation of infrared using a liquid crystal mixture doped with push-pull azobenzene for energy-saving smart windows. ACS Applied Materials & Interfaces 2021, 13, 5028-5033.

(29) Li, K.; Meng, S.; Xia, S.; Ren, X.; Gao, G. Durable and controllable smart windows based on thermochromic hydrogels. ACS Applied Materials & Interfaces 2020, 12, 42193-42201.

(30) Xiao, X.; Shi, D.; Yang, Z.; Yu, Q.; Kaneko, D.; Chen, M. Near infrared-sensitive smart windows from Au nanorod-polymer hybrid photonic hydrogels. New Journal of Chemistry 2021, 45, 4016-4023.

(31) Zhang, L.; Xia, H.; Xia, F.; Du, Y.; Wu, Y.; Gao, Y. Energy-saving smart windows with HPC/PAA hybrid hydrogels as thermochromic materials. ACS Applied Energy Materials 2021, 4, 9783-9791.

(32) Tian, J.; Peng, H.; Du, X.; Wang, H.; Cheng, X.; Du, Z. Hybrid thermochromic microgels based on UCNPs/PNIPAm hydrogel for smart window with enhanced solar modulation. Journal of Alloys and Compounds 2021, 858, 157725.

(34) Sherry, L.; Jin, R.; Mirkin, C.; Schatz, G.; Van Duyne, R. Localized surface plasmon resonance spectroscopy of single silver triangular nanoprisms. Nano Letters 2006, 6, 2060-2065.

(35) Agrawal, A.; Cho, S.; Zandi, O.; Ghosh, S.; Johns, R.; Milliron, D. Localized surface plasmon resonance in semiconductor nanocrystals. Chemical Reviews 2018, 118, 3121-3207.

(36) Kang, Z.; Lee, S. Carbon dots: advances in nanocarbon applications. Nanoscale 2019, 11, 19214-19224.

(37) Mintz, K.; Bartoli, M.; Rovere, M.; Zhou, Y.; Hettiarachchi, S.; Paudyal, S.; Leblanc, R. A deep investigation into the structure of carbon dots. Carbon 2021, 173, 433-447.

(38) Dideikin, A.; Vul', A. Graphene oxide and derivatives: the place in graphene family. Frontiers in Physics 2019, 6, 149.

(39) Wang, Y.; Li, S.; Yang, H.; Luo, J. Progress in the functional modification of graphene/graphene oxide: A review. RSC Advances 2020, 10, 15328-15345.

(40) Farjadian, F.; Abbaspour, S.; Sadatlu, M.; Mirkiani, S.; Ghasemi, A.; Hoseini-Ghahfarokhi, M.; Hamblin, M. Recent developments in graphene and graphene oxide: Properties, synthesis, and modifications: A review. ChemistrySelect 2020, 5, 10200-10219.

(50) Chen, H.; Shao, L.; Ming, T.; Sun, Z.; Zhao, C.; Yang, B.; Wang, J. Understanding the photothermal conversion efficiency of gold nanocrystals. Small 2010, 6, 2272-2280.

(51) Kim, H.; Lee, D. Near-infrared-responsive cancer photothermal and photodynamic therapy using gold nanoparticles. Polymers 2018, 10, 961.

(52) Chen, M.; He, Y.; Huang, J.; Zhu, J. Synthesis and solar photo-thermal conversion of Au, Ag, and Au-Ag blended plasmonic nanoparticles. Energy Conversion and Management 2016, 127, 293-300.

(53) Xiao, J.; Fan, S.; Wang, F.; Sun, L.; Zheng, X.; Yan, C. Porous Pd nanoparticles with high photothermal conversion efficiency for efficient ablation of cancer cells. Nanoscale 2014, 6, 4345-4351.

(54) Tang, S.; Chen, M.; Zheng, N. Multifunctional ultrasmall Pd nanosheets for enhanced near-infrared photothermal therapy and chemotherapy of cancer. Nano Research 2015, 8, 165-174.

(63) Haruta, M.; Daté, M. Advances in the catalysis of Au nanoparticles. Applied Catalysis A: General 2001, 222, 427-437.

(64) Irshad, M.; Arshad, N.; Wang, X. Nanoenabled photothermal materials for clean water production. Global Challenges 2021, 5, 2000055.

(65) Chang, M.; Hou, Z.; Wang, M.; Yang, C.; Wang, R.; Li, F.; Lin Jun. Single-Atom Pd Nanozyme for Ferroptosis-Boosted Mild‐Temperature Photothermal Therapy. Angewandte Chemie 2021, 133, 202101924.

(66) Blemker, M.; Gibbs, S.; Raulerson, E.; Milliron, D.; Roberts, S.; Modulation of the visible absorption and reflection profiles of ITO nanocrystal thin films by plasmon excitation. ACS Photonics 2020, 7, 1188-1196.

(67) Wei, W.; Hong, R.; Jing, M.; Shao, W.; Tao, C.; Zhang, D. Thickness-dependent surface plasmon resonance of ITO nanoparticles for ITO/In-Sn bilayer structure. Nanotechnology 2017, 29, 015705.

(68) Kalanur, S.; Seo, H. Synthesis of CuxS thin films with tunable localized surface plasmon resonances. ChemistrySelect 2018, 3(21), 5920-5926.

(69) Song, G.; Han, L.; Zou, W.; Xiao, Z.; Huang, X.; Qin, Z.; Hu, J. A novel photothermal nanocrystals of Cu7S4 hollow structure for efficient ablation of cancer cells. Nano-Micro Letters 2014, 6, 169-177.

(70) Zhang, J.; Chen, T; Li, X.; Liu, Y.; Liu, Y.; Yang, H. Investigation of localized surface plasmon resonance of TiN nanoparticles in TiNxOy thin films. Optical Materials Express 2016, 6, 2422-2433.

(71) Wang, L.; Zhu, G.; Wang, M.; Yu, W.; Zeng, J.; Yu, X.; Li, Q. Dual plasmonic Au/TiN nanofluids for efficient solar photothermal conversion. Solar Energy 2019, 184, 240-248.

(78) Ding, B.; Shi, M.; Chen, F.; Zhou, R.; Deng, M.; Wang, M.; Chen, H. Shape-controlled syntheses of PbS submicro-/nano-crystals via hydrothermal method. Journal of Crystal Growth 2009, 311, 1533-1538.

(79) Chou, S.; Kaehr, B.; Kim, J.; Foley, B.; De, M.; Hopkins, P.; Dravid, V.; Chemically exfoliated MoS2 as near-infrared photothermal agents. Angewandte Chemie International Edition 2013, 52, 4160-4164.

(80) Wang, X.; He, Y.; Hu, Y.; Jin, G.; Jiang, B.; Huang, Y. Photothermal-conversion-enhanced photocatalytic activity of flower-like CuS superparticles under solar light irradiation. Solar Energy 2018, 170, 586-593.

(81) Xi, M.; Xu, L.; Li, N.; Zhang, S.; Wang, Z. Plasmonic Cu27S24 nanocages for novel solar photothermal nanoink and nanofilm. Nano Research 2022, 15, 3161-3169.

(86) Patsalas, P.; Kalfagiannis, N.; Kassavetis, S. Optical properties and plasmonic performance of titanium nitride. Materials 2015, 8, 3128-3154.

(87) Gschwend, P.; Conti, S.; Kaech, A.; Maake, C.; Pratsinis, S. Silica-coated TiN particles for killing cancer cells. ACS Applied Materials & Interfaces 2019, 11, 22550-22560.

(88) Jiang, W.; Fu, Q.; Wei, H.; Yao, A. TiN nanoparticles: synthesis and application as near-infrared photothermal agents for cancer therapy. Journal of Materials Science 2019, 54, 5743-5756.

(90) Yang, L.; Tong, R.; Wang, Z.; Xia, H. Polydopamine Particle-Filled Shape-Memory Polyurethane Composites with Fast Near-Infrared Light Responsibility. ChemPhysChem 2018, 19, 2052-2057.

(91) Song, M.; Wang, Y.; Liang, X.; Zhang, X.; Zhang, S.; Li, B. Functional materials with self-healing properties: A review. Soft Matter 2019, 15, 6615-6625.

(100) Zhang, J.; Liu, G.; He, F.; Chen, L.; Huang, Y. Au@Cu7S4 yolk-shell nanoparticles as a 980 nm laser-driven photothermal agent with a heat conversion efficiency of 63%. RSC Advances 2015, 5, 87903-87907.

(103) Boden, S.; Bagnall, D. Optimization of moth-eye antireflection schemes for silicon solar cells. Progress in Photovoltaics: Research and Applications 2010, 18, 195-203.

Comment 3:

Please improve the structure of the review paper. For example, the section 2 is entitled as “Types of thermochromic smart windows”. However, there are explanation of some materials. After that, “3 Common photothermal conversion materials” is placed.

Response:

Thanks for Referee’s kind suggestion. After careful consideration, according to the content of the paragraph, we changed the title of section 2 from “Type of thermochromic smart windows” to “Thermochromic Materials for smart windows”.

Comment 4:

In the conclusion section, the authors give suggestion of (1) ~ (3). However, these are typical opinions. Please discuss deeper related to the mentioned references.

Response:

Thanks for Referee’s kind suggestion. According to the Referee’s suggestion, we have discussed deeper related to the mentioned references. The modified is: “For example, in Table 1, the photothermal efficiency of Cu7S4 Nano Superlattice is significantly higher than that of Cu7S4 NPs[99].”, “As Au@Cu7S4 yolk-shell NPs[100], not only retain the respective absorption peaks of Au and Cu7S4 in UV-Vis-NIR spectra, but also show significantly enhanced absorption in the near-infrared region. And the photothermal conversion efficiency is also higher, which shows the total is greater than the sum of parts.”, “In addition to improving the conversion efficiency of photothermal materials, solar absorption efficiency is also an important parameter. In the field of solar cells, bionic anti-reflective periodic structures, such as butterfly wings and moth eyes, are often used on the surface of solar panels[103]. These rough nanostructures can effectively eliminate reflection and help them absorb more sunlight than smooth surfaces. Similarly, this structure is also applicable to smart windows, which will significantly shorten the response time.”

We tried our best to improve the manuscript and made some other changes in the manuscript. These changes will not influence the framework of the paper. And we have highlighted all the changes in the revised manuscript. We appreciate for Editor and Referees’ warm work earnestly, and hope that the correction will meet with approval.

Once again, thank you very much for your comments and suggestions!

Best wishes!

Yours Sincerely,

Haining Ji

Reviewer 2 Report

This paper is a nice and well written review which described several classes of promising photothermal materials. The content of the paper is appropriate and the choice of materials is correct. The language of the paper is fine. The figures are informative. Please, be sure that all of them are original or suitable agreements for reuse are achieved.

I suggest direct acceptance

Author Response

Dear Editor and Referees:

Thanks for your letter and for the Referees’ comments concerning our manuscript entitled “Thermochromic Smart Windows Assisted by Photothermal Nanomaterials” (Manuscript ID: nanomaterials-1989252). We deeply appreciate the time and effort you’ve spent on reviewing our manuscript. These comments are really thoughtful and very helpful for revising and improving our paper, as well as the important guiding significance to our research. During the last days, we have referred to literatures, checked spelling and studied your comments carefully, and we have completed the modifications according to the referees’ comments.

The Referees’ comments and the responses to the comments are as follows:

Reviewer: 2#

This paper is a nice and well written review which described several classes of promising photothermal materials. The content of the paper is appropriate and the choice of materials is correct. The language of the paper is fine. The figures are informative. Please, be sure that all of them are original or suitable agreements for reuse are achieved.

I suggest direct acceptance.

Response:

We thank the Referee for the positive comments on our work.

We tried our best to improve the manuscript and made some other changes in the manuscript. These changes will not influence the framework of the paper. And we have highlighted all the changes in the revised manuscript. We appreciate for Editor and Referees’ warm work earnestly, and hope that the correction will meet with approval.

Once again, thank you very much for your comments and suggestions!

Best wishes!

Yours Sincerely,

Haining Ji

Reviewer 3 Report

This review article reports on the thermochromic smart windows that can regulate their optical properties actively in response to external temperature changes. To overcome the disadvantage of the conventional thermochromic smart windows that have generally problem with high response temperature and low response rate, photothermal nanomaterials that assist the phase transition of the thermochromic materials are investigated. From the overview of the current research, the outlooks and prospects for future research on the phase transition of photothermal materials assisted thermochromic smart windows are suggested in the Conclusions and perspective section. This article is well organized and provides a suggestion to overcome the problem of the conventional thermochromic smart windows with high response temperature and low response rate by improving the photothermal nanomaterials. This topic is suitable for this journal. The comments from this reviewer are as follows,

1.     The caption for Figure 1 should be added more information about (a) (b) (c).

2.     Page 7, line 125-128; “The GO dispersed in the PNIPAm hydrogels can make the originally transparent films reach the LCST at room temperature (30 ℃) and complete the phase change in a short time (Fig. 2b).” It is unclear how short time it changed from Fig. 2b. Please explain in more detail.

3.     Page 11, line 215; Figure 2a is not a proper citation here. Should this be Figure 3d? There is no mention about Figure 3 (a), (b), (d), and (e) in the main text. These figures should be cited in the main text at least once.

4.     Figure 4 (d), please indicate which are at 22°C and which are 36°C.

Author Response

Dear Editor and Referees:

Thanks for your letter and for the Referees’ comments concerning our manuscript entitled “Thermochromic Smart Windows Assisted by Photothermal Nanomaterials” (Manuscript ID: nanomaterials-1989252). We deeply appreciate the time and effort you’ve spent on reviewing our manuscript. These comments are really thoughtful and very helpful for revising and improving our paper, as well as the important guiding significance to our research. During the last days, we have referred to literatures, checked spelling and studied your comments carefully, and we have completed the modifications according to the referees’ comments.

The Referees’ comments and the responses to the comments are as follows:

Reviewer: 3#

This review article reports on the thermochromic smart windows that can regulate their optical properties actively in response to external temperature changes. To overcome the disadvantage of the conventional thermochromic smart windows that have generally problem with high response temperature and low response rate, photothermal nanomaterials that assist the phase transition of the thermochromic materials are investigated. From the overview of the current research, the outlooks and prospects for future research on the phase transition of photothermal materials assisted thermochromic smart windows are suggested in the Conclusions and perspective section. This article is well organized and provides a suggestion to overcome the problem of the conventional thermochromic smart windows with high response temperature and low response rate by improving the photothermal nanomaterials. This topic is suitable for this journal.

Response:

We thank the Referee for the positive comments on our work.

Comment 1:

The caption for Figure 1 should be added more information about (a) (b) (c).

Response:

Thanks for Referee’s kind suggestion. According to the Referee’s suggestion, we have added the information about Figure 1. The added description is: “(a) Metal insulator phase transition (MIT) of vanadium dioxide (VO2).  (b) Liquid crystal changes transmittance by adjusting molecular orientation in response to temperature.  (c) Thermochromic hydrogels change transparency reversibly with temperature.

Comment 2:

Page 7, line 125-128; “The GO dispersed in the PNIPAm hydrogels can make the originally transparent films reach the LCST at room temperature (30 ℃) and complete the phase change in a short time (Fig. 2b).” It is unclear how short time it changed from Fig. 2b. Please explain in more detail.

Response:

Thanks for Referee’s kind suggestion. According to the Referee’s suggestion, we have added a specific time for the film to arrive at LCST. The modified sentence is: “The GO dispersed in the PNIPAm hydrogels can make the originally transparent films reach the LCST at room temperature (30 ℃) and complete the phase change in 2 minutes (Fig. 2b).

Comment 3:

Page 11, line 215; Figure 2a is not a proper citation here. Should this be Figure 3d? There is no mention about Figure 3 (a), (b), (d), and (e) in the main text. These figures should be cited in the main text at least once.

Response:

Thanks for Referee’s kind suggestion. We are very sorry for the incorrect citation. Figure 3 (a), (b), and (e) have been mentioned in the proper place of the text. The modified is: “Its steady-state temperature was close to 69 °C, however, the film without AuNRs only increased by 1 °C under the same conditions (Fig. 3a). And the switching temperature and colors of the film can be flexibly controlled by adjusting the concentration of AuNCs and the kind of thermochromic materials (Fig. 3b).” “When the temperature rises above LCST, the hydrogel structure shrinks and deforms, causing the AgNRs to stand up like flowers (Fig. 3d-e), and its IR emissivity got a slight increase from 0.947 to 0.958.

Comment 4:

Figure 4 (d), please indicate which are at 22°C and which are 36°C.

Response:

Thanks for Referee’s kind suggestion. According to the referee's suggestion, we have indicated which temperature is 22 °C and which temperature is 36 °C in the description, and made marks on the figures. The modified is: “(d) Transparency changes of CsxWO3/PAM-PNIPAM films with different PNIPAM concentrations at 22°C (above) and 36°C (below).

Figure 4. (a) Schematic diagram of the chemical structure and working principle of the photochromic supramolecular hydrogel smart window.[73]  (b) Photographs of ATO composite hydrogel films with EGP5 content of 2, 5, and 8 mol%, respectively, before and after irradiation at 100 mW/cm2 for 10 min.[73]  (c) Transmittance variation of CsxWO3/PAM-PNIPAM film with temperature at 550 nm wavelength.[76]  (d) Transparency changes of CsxWO3/PAM-PNIPAM films with different PNIPAM concentrations at 22°C (above) and 36°C (below).[76]

We tried our best to improve the manuscript and made some other changes in the manuscript. These changes will not influence the framework of the paper. And we have highlighted all the changes in the revised manuscript. We appreciate for Editor and Referees’ warm work earnestly, and hope that the correction will meet with approval.

Once again, thank you very much for your comments and suggestions!

Best wishes!

Yours Sincerely,

Haining Ji

Round 2

Reviewer 1 Report

The authors addreseed my comments well.